# Comparison of Solar Radiation Torque and Power Generation of Deployable Solar Panel Configurations on Nanosatellites

**Syahrim Azhan Ibrahim *** and **Eiki Yamaguchi**

Department of Civil Engineering, Kyushu Institute of Technology, Kitakyushu, Fukuoka 804-8550, Japan; yamaguch@civil.kyutech.ac.jp

*   Correspondence: p595303a@mail.kyutech.jp or syahrim@angkasa.gov.my; Tel.: +81-80-8354-1457

**Abstract:** Nanosatellites, like CubeSat, have begun completing advanced missions that require high power that can be obtained using deployable solar panels. However, a larger solar array area facing the Sun increases the solar radiation torque on the satellite. In this study, we investigated solar radiation torque characteristics resulting from the increased area of solar panels on board the CubeSats. Three common deployable solar panel configurations that are commercially available were introduced and their reference missions were established for the purpose of comparison. The software algorithms used to simulate a variety of orbit scenarios are described in detail and some concerns are highlighted based on the results obtained. The solar power generation of the respective configurations is provided. The findings are useful for nanosatellite developers in predicting the characteristics of solar radiation torques and solar power generation that will be encountered when using various deployable solar panels, thus helping with the selection of a suitable configuration for their design.

**Keywords:** CubeSat; solar radiation torque; deployable solar panel; solar power generation

## 1. Introduction

For satellites in low Earth orbit (LEO), major sources of external disturbances that perturb their total angular momentum include the gravitational field, magnetic field, atmosphere, and solar radiation. At altitudes of 400 to 1000 km, the disturbances that affect those satellites the most are the gravitational and magnetic torques. Solar radiation can also exert an appreciable disturbance torque in which the level could match the magnitude of atmospheric drag at altitudes over 500 km [1]. The order of disturbance magnitude can significantly increase, especially for satellites with large surface areas. Therefore, the effects of solar radiation should also be a concern for nanosatellites, like CubeSats, as their missions have become more advanced and subsequently require more power, which has necessitated the use of deployable solar panels. With the addition of these deployable solar panels, the surfaces of the satellite facing the Sun increase and consequently change the characteristics of the total external disturbances on the satellite.

Deployable solar panels on CubeSat can be used to optimize solar power generation and to accomplish specific missions. Many papers about the former application have been published, wherein novel solar panel configurations were designed and solar energy harvesting was studied [2–5]. For the latter application, in one mission example, the strong atmospheric drag at orbits below 500 km was manipulated to achieve attitude stability [6,7]. The effect of in-orbit temperature variations on the solar cells used by CubeSats has been examined [8], and a structure design was proposed to increase the thermal conductance between the solar array and the solid structure of CubeSats to reduce temperature fluctuations [9].

Most of these CubeSats have their own primary missions, and they rely on commercial suppliers for their deployable solar panels as far as reliability is concerned. Whenever deployable parts are involved, products with a flight heritage record are important for increasing the success rate of their primary missions. Although not explicitly specified, some prematurely failed CubeSats missions have been reported, caused by non-functional power subsystems and mechanical deployment systems [10]. We focused on solar radiation torque resulting from the increased area of solar cells on board the CubeSats. In Section 2, solar radiation pressure is briefly explained. In Section 3, three deployable solar panel configurations that are commercially available are introduced and their reference missions are established for comparison purposes. Formulations applied to obtain the solar radiation torque-related parameters are described. In Section 4, the computation used in simulation software is described in detail. Subsequently, in Section 5, significant results are presented and discussed. The solar power generation of the different configurations is shown. The work here is useful to help CubeSat developers select suitable deployable solar panel configurations for their CubeSats and to foresee possible disturbance patterns that could affect the satellite motion.

## 2. In-Orbit External Disturbances

### 2.1. Solar Radiation Pressure

The solar radiation incident on a spacecraft's surface produces a force that results in a torque about the spacecraft's center of mass. The major sources of solar radiation pressure are (1) solar flux from the Sun, (2) radiation emitted from the Earth and its atmosphere, and (3) solar radiation reflected by the surface and clouds of the Earth, i.e., Earth's albedo. This radiation (photons) contains momentum that creates pressure on the lit surface of the spacecraft. The solar pressure is:

$$p = \frac{S}{c} \tag{1}$$

where $S$ is the mean solar flux at 1367 W/m$^2$ and $c$ is the speed of light. In the vicinity of Earth, the value of $p$ is constant at $4.56 \times 10^{-6}$ N/m$^2$. Earth-emitted radiation can be assumed as uniform over the surface of the earth. The flux is:

$$q = \frac{400}{\left(\frac{r}{R_E}\right)^2} \tag{2}$$

where $r$ is the distance between the Earth's center and the satellite's altitude, and $R_E$ is the earth radius at 6378.165 km. The 400 W/m$^2$ is based on the assumption that the Earth is a black body and has a temperature of 289.8 K. The third source, due to albedo, is complex because the Earth cannot be treated as a point source as the reflectivity varies over the surface. In this work, the reflection from the Earth's surface and from clouds is assumed to be diffuse. The flux due to albedo can then be computed by integrating over the surface of the earth. The equation of the flux is:

$$q = \frac{S}{\left(\frac{r}{R_E}\right)^2} aF \tag{3}$$

where $aF$ is the albedo factor, which is 0.33. Of these three sources, solar flux is the dominant source of solar radiation pressure.

To model the solar radiation forces, incident radiation is assumed to be either absorbed, reflected specularly, reflected diffusely, or some combination thereof. In terms of the fractions of the incoming radiation, the following is true for a surface:

$$\rho_a + \rho_d + \rho_s = 1 \tag{4}$$

where $\rho$ (optical properties) stands for the fraction of photons that are absorbed, diffusely reflected, and specularly reflected. The force caused by solar radiation can be expressed as:

$$\vec{F} = -pA\hat{s}^T n\left(2\left(\rho_s\hat{s}^T n + \frac{\rho_d}{3}\right)n + (\rho_a + \rho_d)\hat{s}\right) \text{ for } \left(\hat{s}^T n\right) > 0 \qquad (5)$$

where $\hat{s}$ is the unit sun vector, $T$ denotes its transpose, $n$ is the unit normal to the surface, and $A$ is the surface's area. From Equation (5), the specular component produces the biggest force, followed by the diffuse component and the absorbed component [11]. $\hat{s}^T n$ is the dot product, which is the cosine of the angle between $\hat{s}$ and $n$. Its positive value means that the surface normal faces the sun direction. Later, we explain that the $\hat{s}$ value is also interchangeable with the other two sources of radiation, which are the Earth's radiation and albedo.

*2.2. Other Disturbances*

To observe how the solar radiation torque level changes due to usage of the deployable solar panels, other major disturbance torques in LEO will also be accounted for in the study. The disturbances consist of the aerodynamics drag, gravity gradient, and the residual dipole.

Aerodynamic force disturbance is due to the interaction between a planetary atmosphere and a spacecraft surface. The disturbance force model used in this study is:

$$\vec{F} = \frac{1}{2}\rho_{atm}C_D A_p \vec{v}\left|\vec{v}\right| \qquad (6)$$

where $\rho_{atm}$ is the atmospheric density, $C_D$ is the drag coefficient, $A_p$ is the projected area, and $\vec{v}$ is the velocity vector. The projected area for a flat plate is $A_p = A\cos\alpha$, where $\alpha$ is the angle between the surface normal and the velocity vector. The value of $\rho_{atm}$ is $3.725 \times 10^{-12} \text{ kg/m}^3$, based on the scale heights atmospheric model [12] (p. 820) whereas the $C_D$ is estimated as 2 [12] (p. 64).

For LEO satellites, which have off-diagonal terms in their inertia matrix, they can experience the gravity gradient disturbance torque from the variation of the Earth's gravitational force. In the case of nadir pointing satellites, the vector torque is modeled as:

$$\vec{T} = 3\omega_o^2\begin{bmatrix} -I_{yz} \\ I_{xz} \\ 0 \end{bmatrix} \qquad (7)$$

where $\omega_o$ is the orbital natural frequency. The solar panels would cause off-diagonal terms, but are very small as they are just flat plates.

The residual dipole disturbance torque results from the interaction of the magnetic field generated by current loops on the spacecraft with the Earth's magnetic field. The torque direction, $T$, is normal to both the satellite's residual dipole, $M$, and the Earth magnetic field vector, $B$, as shown in the equation below:

$$\vec{T} = \vec{M} \times \vec{B} \qquad (8)$$

Both $M$ and $B$ must be resolved into the body frame [11] (p. 137). In this study, the magnetic field model is based on a tilted dipole model [12] (pp. 783–784) whereas a 0.01 $\text{Am}^2$ residual dipole strength along the z-axis would be a good estimation from the literature on CubeSats [13]. The total torque disturbances will be the summation of torques due to the solar radiation pressure, aerodynamics drag, gravity gradient, and the residual dipole.

**3. Model Parameters**

In this section, the satellite surface parameters that consist of the unit normal to the surface, area of the surface, and the surface optical properties mentioned in Equation (5) are described. Then,

the formulations used to determine the unit sun vector are shown. Since an improved solar power generation is always the main purpose for using a deployable solar array, the method to compute solar power generation is also presented so that the results can be compared.

### 3.1. Satellite Configuration

A CubeSat has a standard built dimension of 10 cm³ for one unit (1U) size with no protuberant parts at launch. A 3U CubeSat is basically composed of three 1U CubeSats stacked lengthwise. For the present analysis, three 3U CubeSat models were configured with deployable solar panels as depicted in Figure 1. The models use a simplified model of areas and normal, and consist of a set of vertices and faces defining the exterior of the satellites. The optical property of the body surfaces that are mounted with solar cells is assigned as a solar cell, whereas the surfaces that have no solar cells are assigned as radiator surface. All three models can be considered as common designs as they are commercially available [14–16] and similar designs were used for some CubeSat missions and studies [2,5–7,17,18].

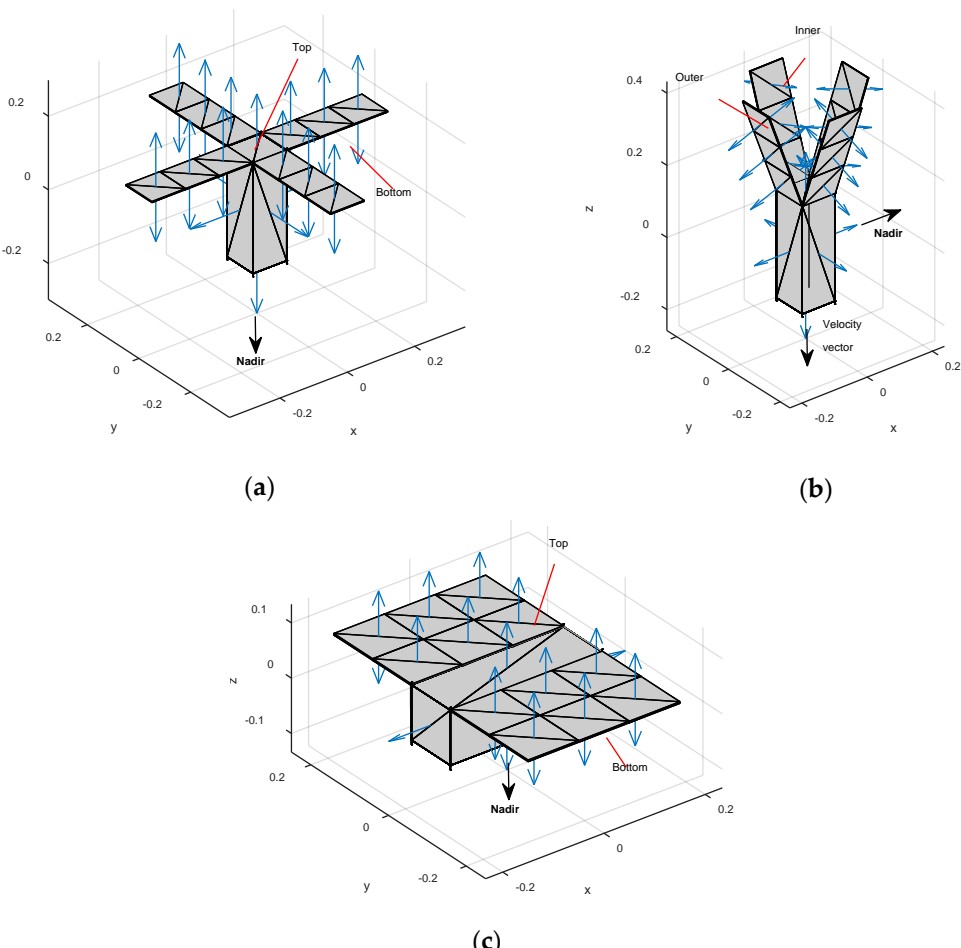

**Figure 1.** CubeSat geometry of (**a**) model 1, (**b**) model 2, and (**c**) model 3 with the surface normal indicated by the solid arrows.

The cube faces have six surfaces, which create the satellite body. Each model had four fixed, body-mounted solar panels on the long section surfaces. Each deployable solar panel had a length of 300 mm, a width of 100 mm, and a thickness of 3 mm. Each panel is divided into three parts with each part having two surfaces: Top and bottom. Therefore, each deployable solar array has six surfaces that can be defined differently. Altogether, each model has 30 surfaces to be defined. Surface normal

vectors are shown by the blue arrows in Figure 1. In the simulation code, both areas and surface normal vectors are gathered into their respective matrix and placed into the *x*-, *y*-, or *z*-axis.

For ease of comparison, all the CubeSat models used have the same mission: A nadir-pointing mission. We assumed that their respective attitude control systems are able to maintain the pointing in the directions assigned. The state-of-the-art CubeSat technology has been found to be potentially compatible with some Earth observation missions [19]. For model 1, the satellite has four deployable solar panels attached 90 degrees at the +*z*-axis short edges of the satellite body. The top surfaces of the solar panels and four sides of the *x*-axis and *y*-axis satellite body are assigned as solar cells, whereas the bottom sides of the extendable panels and *z*-axis sides of the body are assigned as a radiator. The nadir pointing is on the −*z*-axis of the satellite body. Model 2 has its extendable solar panels angle deployed at fixed 30 degrees with respect to the +*z* axis of the satellite. This configuration is normally known as the space-dart configuration. The outer sides of the deployable solar panels are assigned solar cell surface properties, whereas the inner sides are assigned radiator surface properties. The remaining surfaces of the main satellite body part are assigned as model 1. This configuration was found to be capable in providing attitude aerodynamic stability for CubeSats that orbit at altitudes below 500 km [6,7]. To manipulate the aerodynamic drag for attitude stabilization, the short section of the satellite body that does not have deployable solar panels (in this case, the −*z*-axis side) point in the direction of the velocity of the satellite. We set one long section of the body to be fixed to the nadir pointing direction (in this case, the +*x*-axis side) to match the nadir pointing missions of the other two models to facilitate the solar radiation torque comparison study. Lastly, model 3 has two-double solar panels deployed along the long edge of the 3U CubeSat body. Like model 1, the top surfaces of the solar panels are solar cells and the bottom sides are radiators. The nadir pointing is also on the −*z*-axis of the satellite body.

*3.2. Position of the Sun and Eclipse Condition*

To compute the Sun's position with respect to the satellite for any given location at a given time, there are three steps to follow [20]: (1) Calculate the Sun's position in the ecliptic coordinate system, (2) convert the result from step 1 to the Earth-centered inertial (ECI) frame, and (3) convert the result from step 2 to the satellite body coordinate system.

For the first step, a simple algorithm is used to generate the position of the Sun in the ecliptic coordinate system to a precision of about 1 arcminute for dates between 1950 and 2050. First, the number of days, *j*, is computed from the epoch referred to as the Julian date, 2,451,545.0:

$$j = JD - 2451545.0 \qquad (9)$$

where *JD* is the Julian date of interest. Then, the mean longitude of the Sun (*L*), mean anomaly of the Sun (*g*), and ecliptic longitude of the sun (*λ*) are computed:

$$L = 280.460° + 0.9856474°j \qquad (10)$$

$$g = 357.528° + 0.9856474°j \qquad (11)$$

$$\lambda = L + 1.915° sing + 0.020° sin2g \qquad (12)$$

where all the values are in the range of 0 to 360°. The distance of the Sun from the Earth ($u_{sun}$) in the unit meter can be approximated as follows:

$$u_{sun} = (1.00014 - 0.01671 cosg - 0.00014 cos2g) * 149600e3 \qquad (13)$$

Other than $\lambda$ and $u_{sun}$, another parameter to form a complete position of the sun in the ECI frame is the ecliptic latitude, $\beta$. The Sun's ecliptic latitude can be approximated by $\beta = 0$. Next, in step 2, the unit sun vector ($\hat{u}_{sun}$) in the equatorial coordinate system is computed as follows:

$$\hat{u}_{sun} = \begin{bmatrix} cos\lambda \\ cos\epsilon \; sin\lambda \\ sin\epsilon \; sin\lambda \end{bmatrix} \tag{14}$$

where $\epsilon$ is the obliquity of the ecliptic, which can be approximated by:

$$\epsilon = 23.439° - 4.00 \times 10^{-7} j \tag{15}$$

In step 3, the $\hat{u}_{sun}$ must be converted to a unit vector from the satellite toward the Sun ($\hat{s}$), to be used in Equation (5). In a later section, this conversion will be explained.

Next, the Sun's distance and its position from the Earth can be used to determine the conditions under which eclipses occur. The problem of calculating the eclipse times of a spacecraft orbiting the Earth has been studied in depth using various methods [12,21–23]. In this work, the existing spherical Earth conical shadow model given by Wertz [12] was used—the atmospheric effects are neglected. We were only concerned about the eclipse seen by objects of negligible sizes, such as satellites. The variables for eclipse geometry are shown in Figure 2.

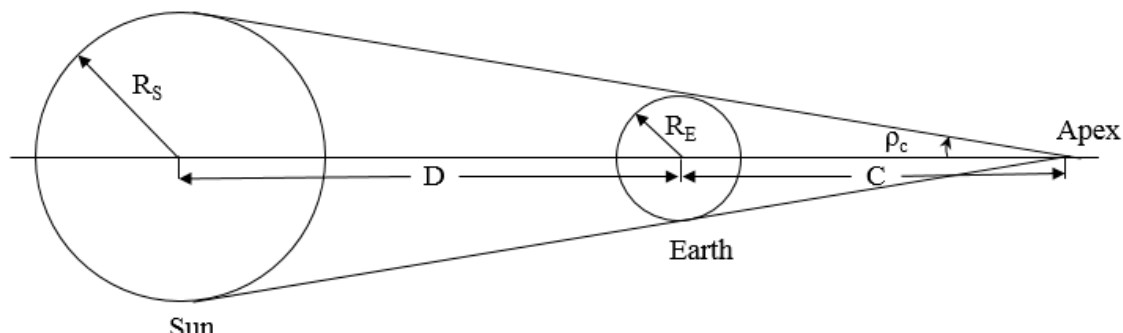

**Figure 2.** Variables for eclipse geometry, where $D$ is the distance from the Earth to the Sun, $R_E$ is the radius of the Earth, and $R_S$ is the radius of the visible surface of the Sun. By simple trigonometry, the distance from the center of the Earth to the apex of the shadow cone, $C = 1.385 \times 10^6$ km and $\rho_c = 0.264°$.

To find the condition when a satellite is in eclipse condition, let $\vec{D}_S$ be the vector from the satellite to the Sun and $\vec{D}_E$ be the vector from the satellite to the center of the Earth. Referring to Figure 3, from the satellite's position, the parameters to be determined are the angular radius of the sun, $\psi_S$, the angular radius of the Earth, $\psi_E$, and the angular separation, $\theta$, between the Sun and the Earth. These three parameters are given by:

$$\psi_S = sin^{-1} \frac{R_S}{D_S} \tag{16}$$

$$\psi_E = sin^{-1} \frac{R_E}{D_E} \tag{17}$$

$$\theta = cos^{-1}\left(\hat{D}_S \cdot \hat{D}_E\right) \tag{18}$$

The necessary conditions for the total eclipse occur when:

$$D < D_S < (D + C) \text{ and } (\psi_E - \psi_s) > \theta \tag{19}$$

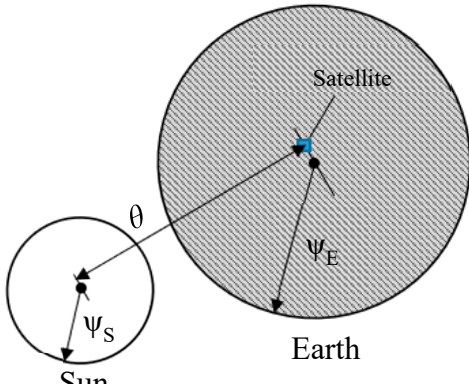

**Figure 3.** Solar eclipse geometry. The parameters to be determined are the angular radius of the Sun, $\psi_S$, the angular radius of the Earth, $\psi_E$, and the angular separation, $\theta$, between the Sun and the Earth.

*3.3. Solar Power Calculation*

The power produced by the solar arrays mounted on the body and the deployable solar panels for any given location at a given time is [11]:

$$Power = \sum_{i=1}^{16} \eta \rho_a A_i Sn_i \hat{s}^T n \ for \left( \hat{s}^T n \right) > 0 \tag{20}$$

where $i$ is the solar cell number, $\eta$ is the solar cell efficiency, and $\hat{s}^T n$ is the dot product, equivalent to the cosine of the angle between $\hat{s}$ and $n$. Whenever the resultant dot product is negative, it means that the particular surface does not face the Sun, and hence does not produce any power. In addition, since the solar cell mounted is normally smaller than the panel area, we set the $A$ value in Equation (20) to 70% of the total panel surface area, for a more realistic result.

## 4. Simulation Program Flow

The simulation software program was implemented using Princeton Satellite Systems CubeSat Toolbox [24], which is a MATLAB® Toolbox (The MathWorks Inc., Natick, MA, USA) for designing CubeSats and analyzing CubeSats missions. Figure 4 shows the flow chart of the simulation software program. Only solar radiation torque computation will be explained since the same procedure is applied to other disturbances albeit with differences in the environment parameters. The first step is to define data for the CubeSat model. Data that are populated consist of the satellite's center of mass $(C_m)$, deployable solar panels dimension and configuration, surface vectors with respect to the origin $(rFace)$, surface area $(A)$, surface the outward unit normal $(n)$, and the optical properties $(\rho)$ of each surface. In the second step, simulation time is first set as a reference to pre-allocate the maximum amount of space required by arrays of data in the subsequent process. This can improve the code execution time.

Next, the initial position and velocity vectors of the satellite are propagated based on defined Keplerian elements. An orbit can be described using six classical Keplerian orbital elements: Semi-major axis $(a)$, eccentricity $(e)$, right ascension of the ascending node $(RAAN)$, inclination $(i)$, argument of perigee, and mean anomaly. These elements can provide information on the orbit size, orbital plane, and the position of the satellite. However, it is easier to use Cartesian elements that consist of a position, $\vec{r}$, and velocity, $\vec{v}$, for propagating the orbit. By using small step sizes, a numerical integrator can determine the new position and the velocity of the satellite given the current position, velocity, and acceleration. Subsequently, acceleration from other forces can be added, such as the solar radiation pressure effect. The Cartesian coordinate frame used in step 2 is the Earth-centered inertial (ECI) coordinate frame, which has its origin at the center of the Earth and is inertially fixed (Figure 5). The fundamental plane contains the equator, and the positive $X$-axis $(X_I)$ points in the vernal equinox direction. The $Z$-axis $(Z_I)$ points in the direction of the geographical north pole and the $Y$-axis $(Y_I)$

completes the right-hand set of coordinate axes. The algorithm to convert between Kepler elements to Cartesian elements are well documented, for example, by Wertz [12] and Sidi [25].

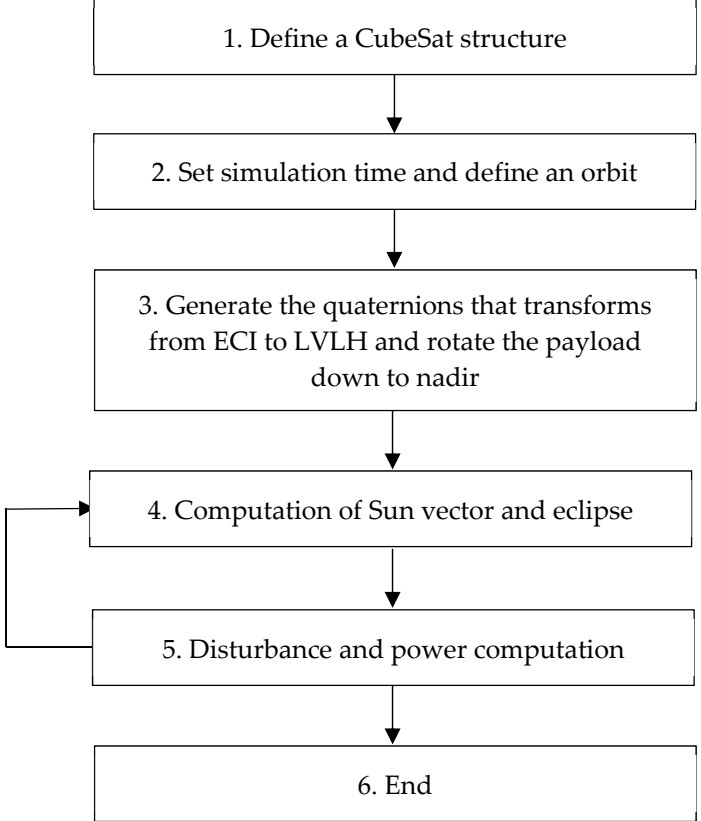

**Figure 4.** Flowchart of disturbance computation.

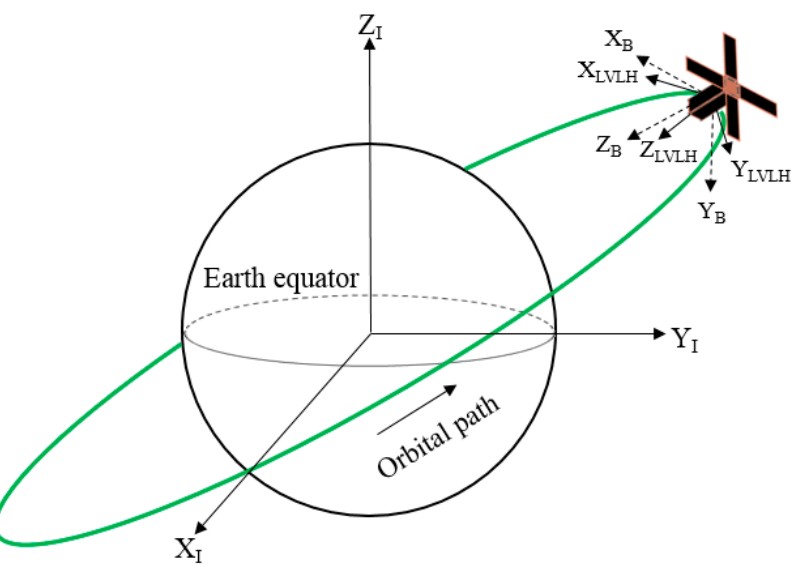

**Figure 5.** Coordinate frames for the nadir-pointing satellite.

Although the simulator is able to consider various orbital parameters, some simplifications can be applied when studying the solar radiation torque in LEO. Firstly, since the amount of momentum flux that is incident on an earth-orbiting satellite is roughly the same within the earth's vicinity (i.e., at 1 astronomical unit) [1], the altitude or the semi-major axis can be fixed at one value. Then, the focus is

on the shape of the disturbance during the transition from eclipse to daylight and vice versa. The rapid changes in thermal loading initiated during the transitions have been known to cause thermally induced dynamics in satellite appendages, such as the deployable solar arrays [26–28]. Therefore, a class of orbits can be selected, such as the orbit occupied by the International Space Station (ISS). The orbit is essentially important for CubeSats since, from 2012 to 2018, more than 200 CubeSats have been launched from ISS, mainly through services provided by the Japan Aerospace Exploration Agency (JAXA) [29] and the National Aeronautics and Space Administration (NASA) [30]. Next, a circular orbit is assumed; therefore, eccentricity is always zero and the argument of perigee can be removed. This is sensible since most satellites in LEO have an eccentricity less than 0.01 [5]. The inclination (*i*) and *RAAN* define the orbit plane, which means that the Sun's angle to the satellite depends on them. Therefore, the values of *i* and *RAAN* vary throughout the simulations.

In the third step, the satellite is aligned to the local vertical local horizontal (LVLH) coordinate frame from the ECI frame. The LVLH is another coordinate frame that has its origin at the center of the satellite and fixes to the orbit (Figure 5). It is commonly assigned to Earth/nadir-pointing satellites, where $Z_{LVLH}$ points towards the Earth, $Y_{LVLH}$ is normal to the local plane with a negative direction, and $X_{LVLH}$ completes the right-handed orthogonal axis set. By completing the transformation from ECI to LVLH, the motion of the satellite can be described by the translational motion of the center of mass of the satellite around the center of the mass of the Earth. The transformation from ECI to LVLH can be performed following the procedure outlined by Paluszek et al. [11]. Let the ECI unit vector be $\hat{u}_I$, and let $[\ ]_u$ represent the unit operator on a vector, returning a vector of the same direction with length 1:

$$\hat{u}_I = M^T \hat{u}_{LVLH} \tag{21}$$

where *M* is a transformation matrix used to transform vectors from the ECI frame to the LVLH frame and $\hat{u}_{LVLH}$ is a unit vector defined in the LVLH frame.

The transformation matrix, *M*, may be computed from the spacecraft ECI position $\left(\vec{r}\right)$ and velocity $\left(\vec{v}\right)$ vectors as follows:

$$M = \begin{bmatrix} x^T \\ y^T \\ z^T \end{bmatrix} \tag{22}$$

where $y = \left[\vec{v} \times \vec{r}\right]_u$, $z = -\left[\vec{r}\right]_u$, and $x = [y \times z]_u$. The transformation matrix above has nine elements, which complicates its application for the propagation of an object. A more efficient method involves using quaternion terminology, which has a mere four elements. The quaternion (in this case, the quaternion from the ECI to LVLH frame) is represented by a four-row vector:

$$q_{I/LVLH} = \begin{bmatrix} q_0 \\ q_1 \\ q_2 \\ q_3 \end{bmatrix} = q_0 + iq_1 + jq_2 + kq_3 \tag{23}$$

where $q_0$ is a scalar component and $q_{1-3}$ are vector components. The conversion of the transformation matrix to quaternion can be obtained using the procedure outlined by Sidi [25].

Next, we wanted to rotate the payload down to the nadir direction. As an example, for the model 1 CubeSat, the nadir pointing is at the −*z*-axis. The pointing target vector can be achieved by rotating the CubeSat 180° around the *x*-axis. The quaternion of this rotation (from the LVLH coordinate frame to the body coordinate frame), herewith named $q_{LVLH/B}$, is:

$$q_{LVLH/B} = cos\theta + i(xsin\theta) + j(ysin\theta) + k(zsin\theta) \tag{24}$$

where $\theta$ is the angle of rotation and $x$, $y$, and $z$ are vectors representing the rotation axes. Finally, the transformation from the ECI frame to the body frame can be obtained as follows:

$$q_{I/B} = q_{I/LVLH}q_{LVLH/B} \tag{25}$$

From here on, each set quaternion and its correspondence time and other properties defined in the first step are processed one at a time in the fourth and fifth steps.

In step 4, the Sun distance, its unit vector in the ECI frame, and the position when the satellite is in daylight or eclipse are computed using the methods explained in Section 3.2. In the last step, solar pressure force due to the solar flux from the Sun, albedo, and Earth-emitted radiation in the satellite body frame are collected, and then the disturbance torques are calculated. For the surfaces that face the solar flux, to find the unit Sun vector, $\hat{s}$, in Equation (5), the unit Sun vector, $\hat{u}_{sun}$, obtained in Equation (10) must be converted to the unit vector from the satellite toward the Sun. This can be completed using quaternion transformation as follows:

$$\hat{s} = q_{I/B}\hat{u}_{sun}q^*_{I/B} \tag{26}$$

where $q^*_{I/B}$ is the conjugate of $q_{I/B}$. For the surface not facing the Sun, the $\hat{s}$ is replaced as follows:

$$\hat{s} = q_{I/B}\left(-\left[\vec{r}\right]_u\right)q^*_{I/B} \tag{27}$$

The value of $\hat{s}$ in Equation (23) will be used in solar flux force and solar power calculations, whereas the value of $\hat{s}$ in Equation (24) will be used for both albedo force and Earth-emitted radiation force calculation. The negative sign in Equation (27) indicates that the source of radiation is from the nadir direction. The total force on the satellite body, $\vec{F}_B$, will be the summation of these three forces. The eclipse condition determined in the fourth step is also considered during the calculation. The force on a surface acts as a torque on the satellite body if the face is offset from the center of mass, as:

$$\vec{T} = \sum\left(\vec{rFace} - C_m\right) \times \vec{F}_B \tag{28}$$

## 5. Disturbance and Power Evaluation

A series of simulations were conducted to evaluate the solar radiation disturbance on the CubeSat models in LEO. The data of this orbit are shown in Table 1. The orbit used resembles the orbit occupied by the ISS. The simulation time was set to one year to demonstrate the effects of season change. One year is a reasonable period since most CubeSats are currently designated with a mission lifetime less than one year long. The orbital parameters were simplified to focus solely on the characteristics of solar radiation torque. Therefore, the variation in inclination from 0 to 90 degrees is shown to demonstrate the variation in solar radiation torque due to satellite orientation with respect to the Sun. The fluctuation range of the disturbance torques between positive and negative values are used to compare the performance among the models tested. Additionally, the solar power generation of the different configurations was compared. Details of the satellite specifications and disturbance parameters are provided in Tables 2 and 3.

The simulation time of the six orbits of the eclipse fraction at 408 km altitude is depicted in Figure 6. In each orbit period, the eclipse fraction lasts about 2180 s, whereas the remaining daylight period lasts around 3352 s. Accordingly, the fraction of solar flux follows suit (i.e., occurs only during the daylight period). As shown in Figure 6, the Earth-emitted radiation is constant throughout the orbit, since we previously assumed that it is uniform over the surface of the earth, whereas the reflected radiation due to albedo varies with the Sun vector and satellite vector from the Earth.

**Table 1.** Orbital parameters.

| Parameter | Value |
|---|---|
| Semi major axis (*a*) | 408 km |
| Orbit inclination (*i*) | 51.64° |
| Initial right ascension of the ascending node (RAAN) | 0° |
| Argument of perigee (*ω*) | 0° |
| Eccentricity (*e*) | 0 |
| Initial mean anomaly (*M*) | 0° |
| Initial Julian date | 2,458,563 |

**Table 2.** CubeSats specification.

| Item | Specification | Value |
|---|---|---|
| All three models | Center of mass (*x, y, z*) | (0, 0, 0) |
| | Residual dipole (*x, y, z*) | (0, 0, 0.002 Am$^2$) |
| Solar cell surface properties [24] | Absorbed ($\rho_a$) | 0.75 |
| | Diffuse ($\rho_d$) | 0.08 |
| | Specular ($\rho_s$) | 0.17 |
| Radiator surface properties [24] | Absorbed ($\rho_a$) | 0.15 |
| | Diffuse ($\rho_d$) | 0.16 |
| | Specular ($\rho_s$) | 0.69 |

**Table 3.** Disturbance parameters.

| Parameter | Value |
|---|---|
| Solar flux (*S*) | 1367 Wm$^{-2}$ |
| Earth radiation | 400 Wm$^{-2}$ |
| Albedo factor (*aF*) | 0.33 |
| Atmospheric density ($\rho_{atm}$) | $3.725 \times 10^{-12}$ kg/m$^3$ |
| Drag coefficient ($C_D$) | 2 |

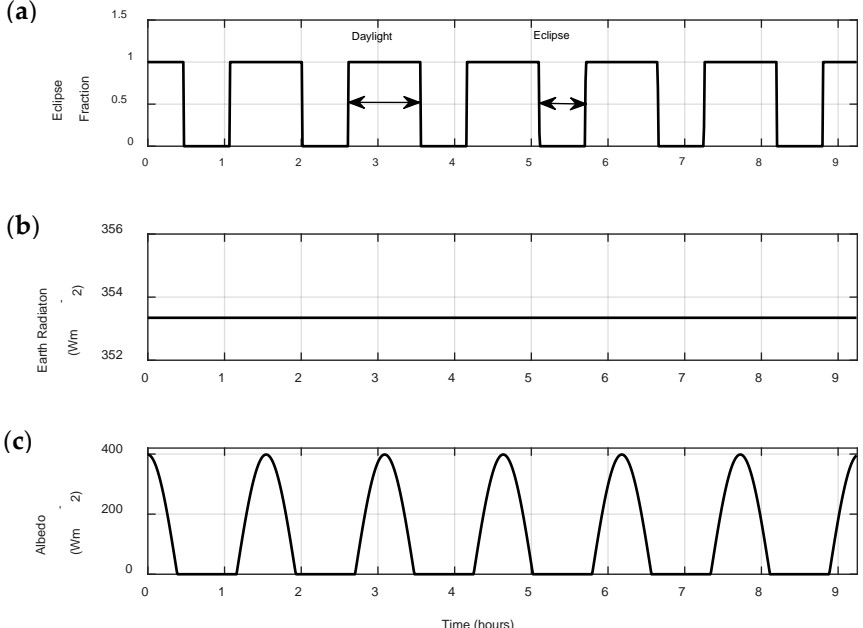

**Figure 6.** (**a**) Eclipse fractions, (**b**) earth radiation, and (**c**) radiation due to albedo at a 408 km altitude.

### 5.1. Solar Radiation Torque on Model 1

The simulation results of the solar radiation torque in model 1 in the *x*-axis ($T_x$) and *y*-axis ($T_y$) are presented in Figure 7. The results show the one-year orbit period with the start of seasons indicated, whereas sub-figures provide more detailed disturbance characteristic of the six orbits' simulation time. At the inclination set, the *RAAN* varies over 360° in a one-year simulation. There is no net torque in the *z*-axis ($T_z$), since the deployable solar panels form a symmetric satellite configuration with respect to the solar forces acting on it. Differences in the disturbance torque level due to seasonal effects can be seen as the Earth's axis tilts with respect to the Sun's rays changing. These effects are more apparent on $T_x$ than $T_y$. During the summer solstice and winter solstice, the $T_x$ is at a maximum and drops to almost zero during the autumnal equinox and vernal equinox. $T_y$ registers its maximum and minimum in opposite seasons. Figure 7a shows the level of $T_x$ during the summer solstice. The disturbance is mainly caused by the solar flux pressure on the deployable solar panels as they face the Sun throughout the daylight period. The level slowly increases when the satellite leaves the eclipse, and reaches its peak when the satellite passes through the equatorial line and subsequently declining until it enters the eclipse. However, the more concerning characteristic is the sudden torque increase and decrease at the edge of the eclipse, since the rapid change in the disturbance torque might cause sudden rotational motion of the satellite. For this particular case, the condition occurs because when the satellite is in the transition from eclipse to daylight and vice versa, two different sides exchange positions to face the solar flux from the Sun. For example, when the satellite emerges from eclipse, the solar flux first hits the solar cell surface on its body for a while before the deployable solar panels face the Sun. Figure 7b shows the level of $T_y$ during autumnal equinox, at which the $T_y$ is at maximum. As in $T_x$, sudden torque changes can be seen during eclipse transitions. During the day, $T_y$ has two peaks due to differences in the position vector of the Sun when the satellite is in the southern and northern hemispheres of the Earth.

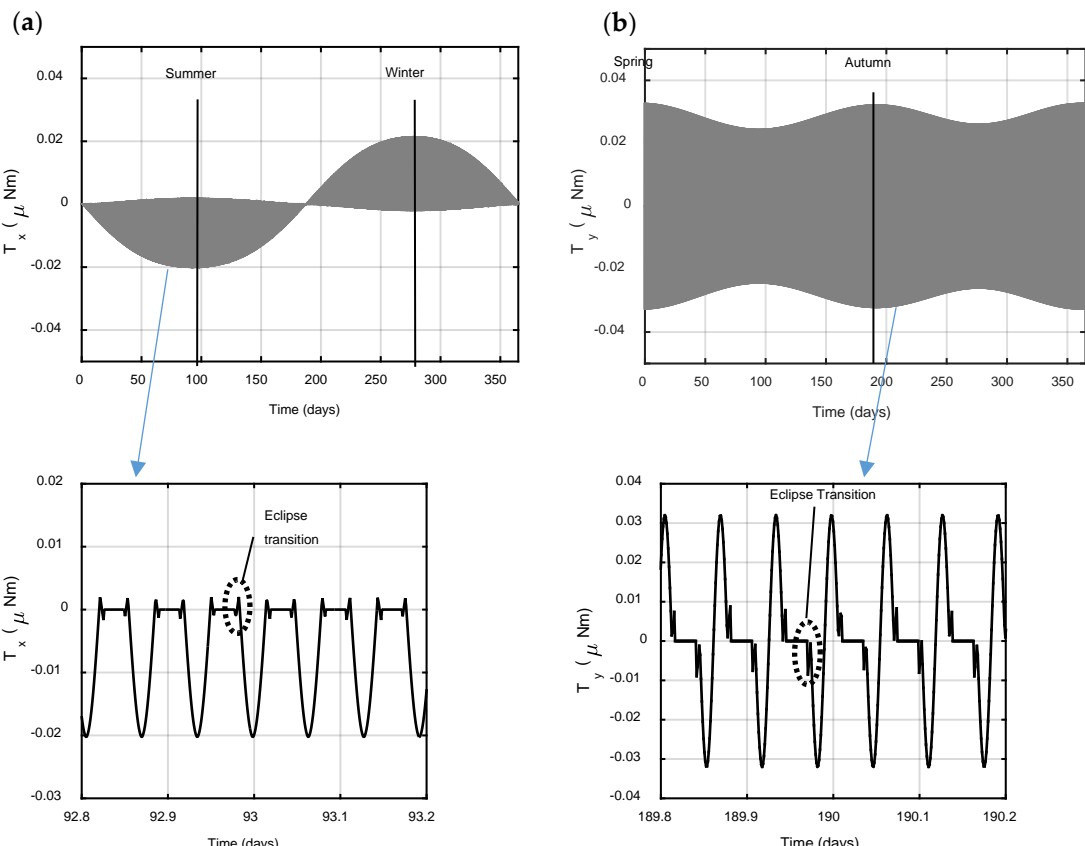

**Figure 7.** Solar radiation torques in model 1 in the International Space Station (ISS) orbit for one-year simulation and six orbit periods at seasonal start time: (**a**) $T_x$ and (**b**) $T_y$.

Next, the shapes of solar radiation torques with inclination angles varying from 0° to 90° are depicted in the three-dimensional (3D) plots of Figure 8. Each plot is an orbit long simulation taken during the summer solstice. The surface plots show the time and angle at which the torque magnitudes are among the maximum at their respective axes. In Figure 8a, the $T_x$ changes direction at the inclination of 23.4° due to the Earth's obliquity. The maximum $T_x$ occurs in the region when the inclination is around 70°. The shape of $T_y$ with varying inclination angles is shown in Figure 8b. The overall shape of the plot is basically the same regardless of the inclination, but the levels differ accordingly. From the 0° inclination, its maximum level increases until the highest, which occurs when the inclination is 23.4°, then subsequently declining almost linearly with increased inclination.

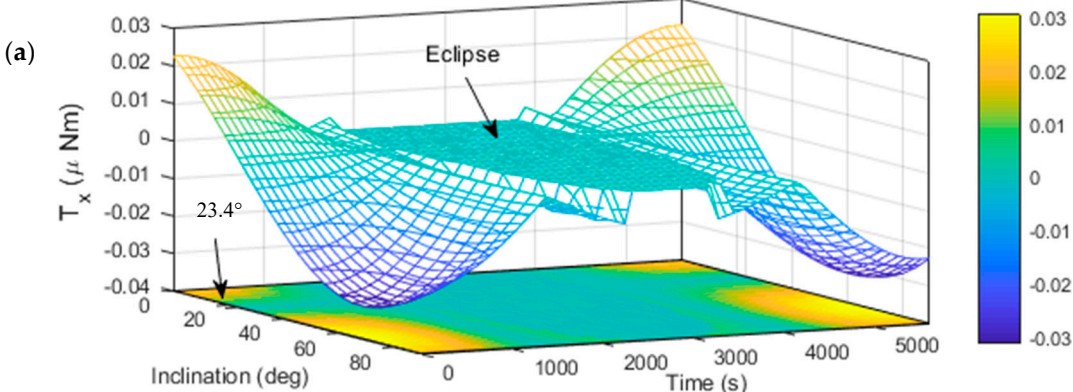

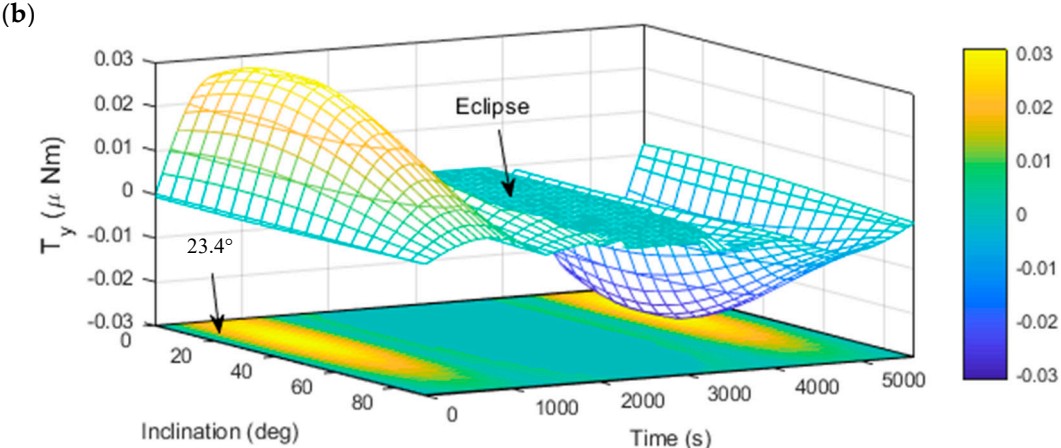

**Figure 8.** One orbit period of solar radiation torques in model 1 in ISS orbit with varying inclination angles. (**a**) $T_x$ and (**b**) $T_y$.

## 5.2. Solar Radiation Torque in Model 2

The simulation results of the solar radiation torque in model 2 are presented in Figure 9. The shape of the one-year long simulation, $T_x$, is the opposite of model 1, whereas $T_y$, similar to model 1, always fluctuates between positive and negative torques throughout the orbit. Compared to model 1, the fluctuation range of $T_x$ and $T_y$ increased by 134% and 108%, respectively. The sub-figures detailing the shape of the torques in the six orbit periods further reveal the differences in comparison to model 1. The torques produced during the eclipse transitions look rather smooth compared to those in model 1, which occurred due to differences in the solar panel configurations and the satellite's velocity direction, although the vector direction of the Sun to the model 2 satellite is different, resulting in the occurrence of $T_z$ as well, where the order of magnitude is 10 times lower than $T_x$.

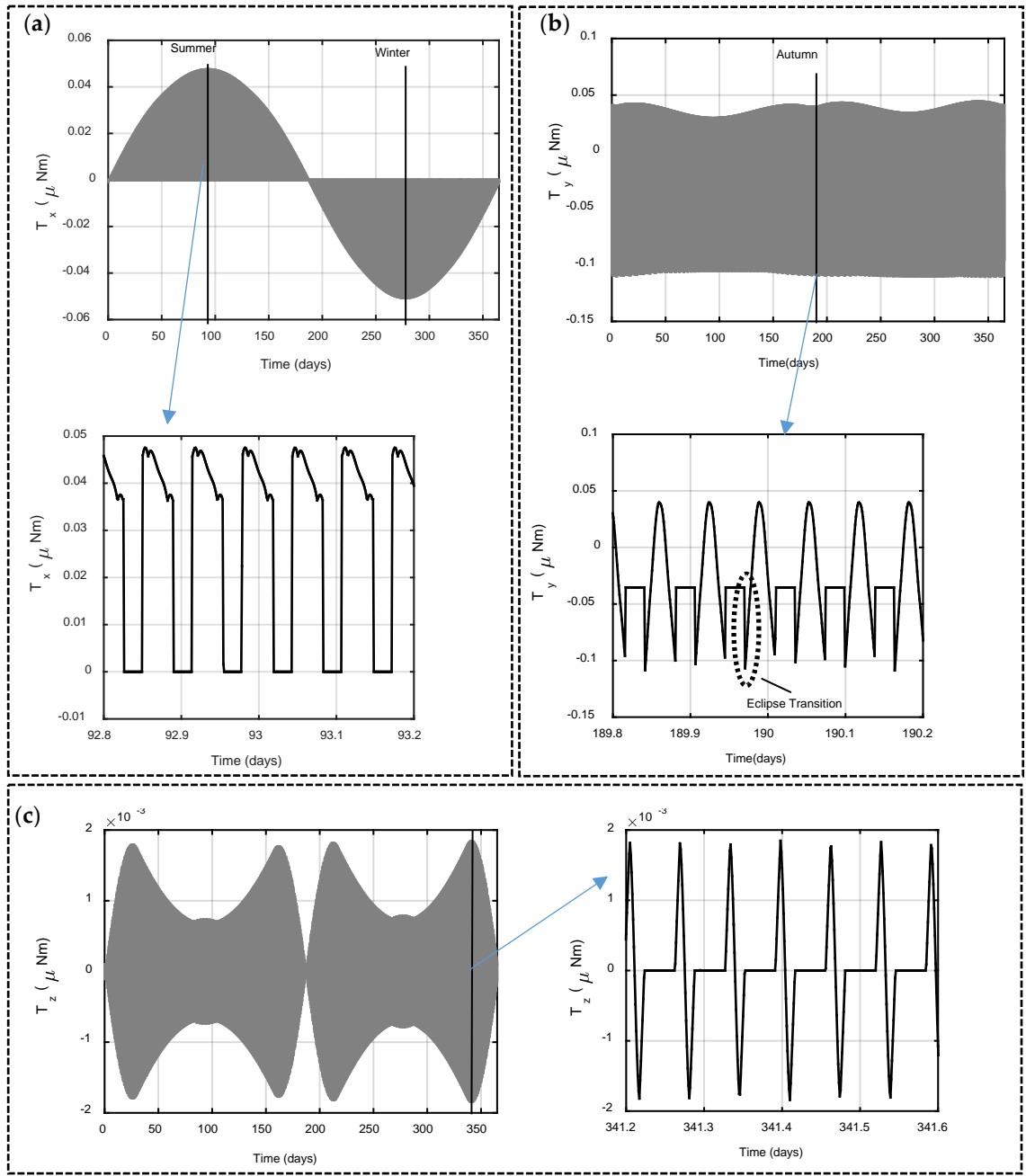

**Figure 9.** Solar radiation torques in model 2 in ISS orbit for one-year simulation and six orbit periods at seasonal start time: (**a**) $T_x$, (**b**) $T_y$, and (**c**) $T_z$.

From the 3D torque plots during the summer solstice in Figure 10, the inclination angles would be a factor of the level of $T_x$. The higher the inclination, the higher the magnitude of $T_x$. For $T_y$, regardless of the inclination angle, maximum levels always occur during eclipse transitions. The effects of the inclination angle change can also be observed in the pattern of $T_z$. Four peaks occur in high inclination, but only two peaks in low and mid inclinations.

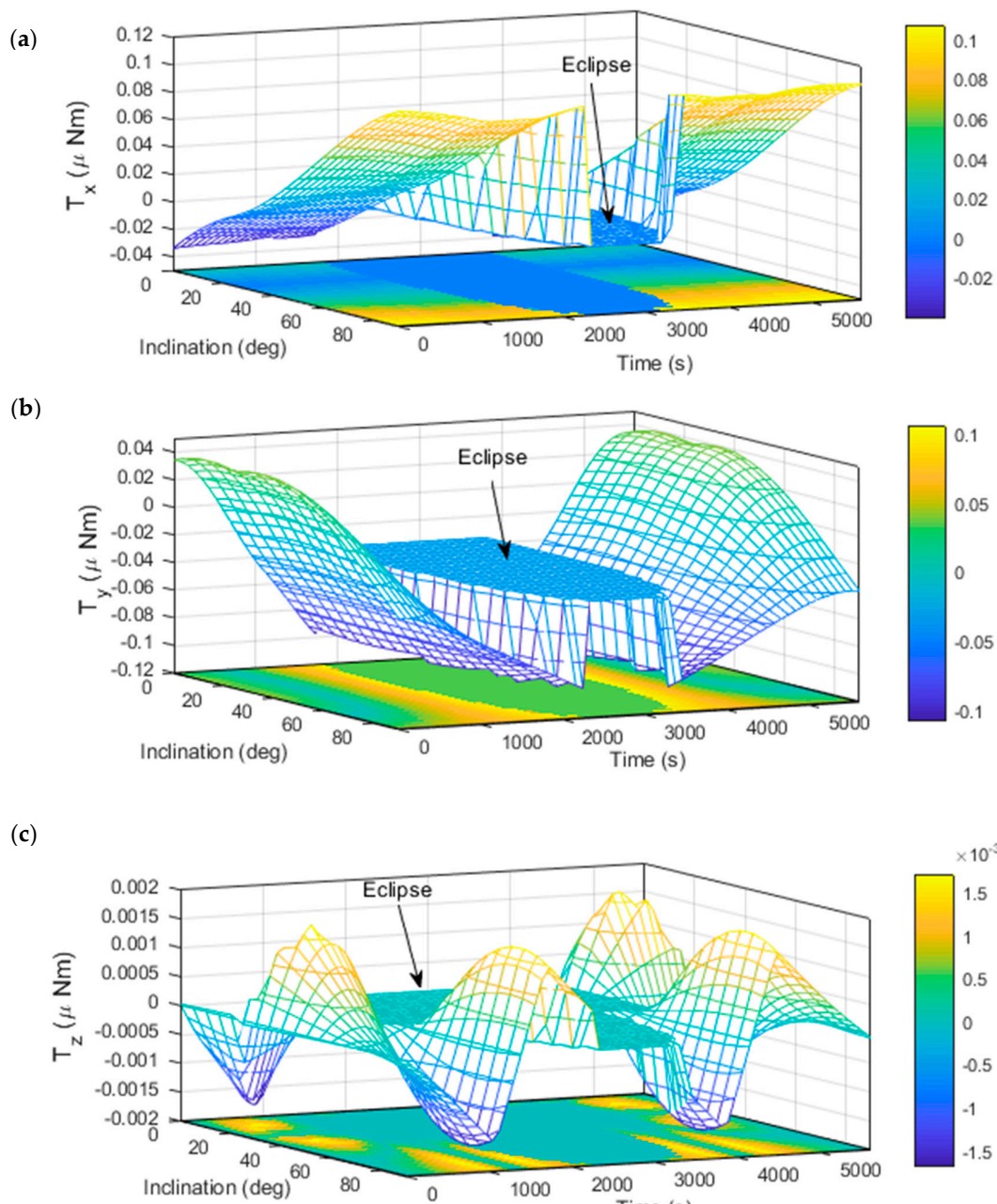

**Figure 10.** One orbit period of solar radiation torques in model 2 in ISS orbit with varying inclination angles: (**a**) $T_x$, (**b**) $T_y$, and (**c**) $T_z$.

### 5.3. Solar Radiation Torque in Model 3

The simulation results of the solar radiation torque in model 3 are presented in Figure 11. The shapes of solar radiation torques, taken during the summer solstice with the inclination varying from 0° to 90° are depicted in 3D plots in Figure 12. As expected, results of model 3 are similar to those of model 1 because the vector direction of the Sun to the deployable solar panels and the pointing mission are similar to model 1. The main difference is the lower peak torques in both $T_x$ and $T_y$. This was expected since the distances between solar panels and the satellite body of model 3 are shorter than in model 1. Compared to model 1, the fluctuation ranges of $T_x$ and $T_y$ decreased by 62% and 70%, respectively.

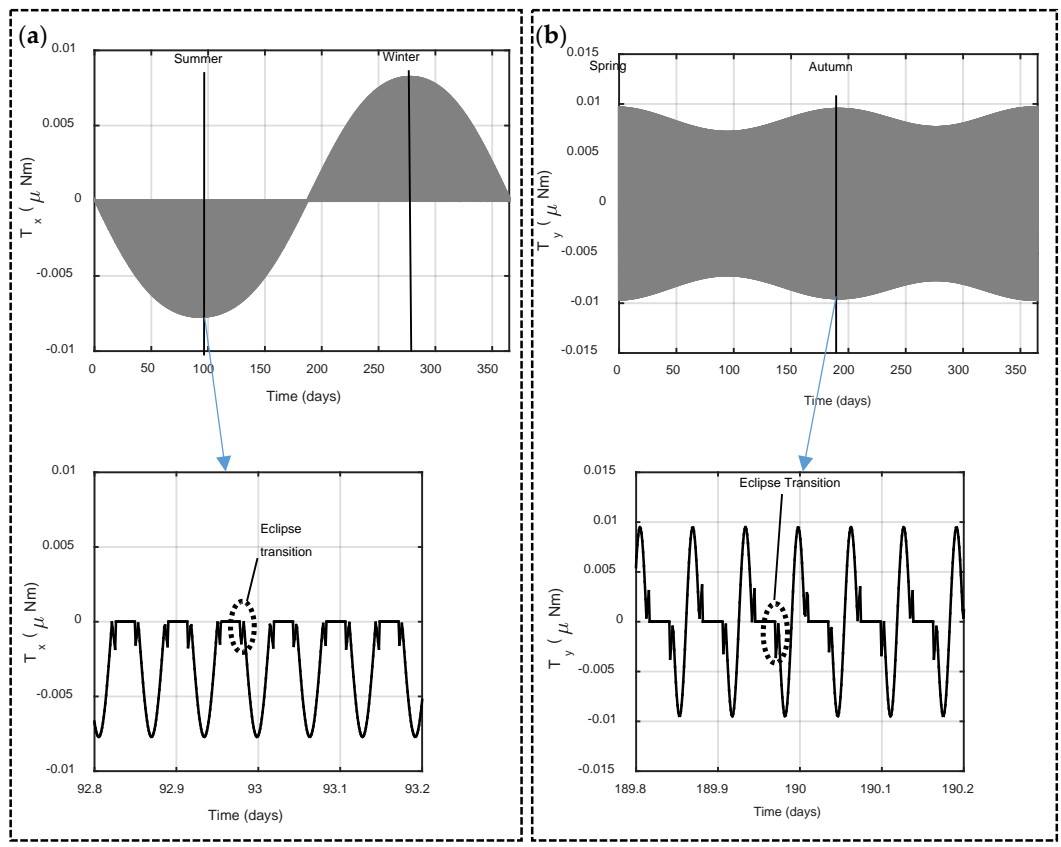

**Figure 11.** Solar radiation torques in model 3 in ISS orbit for one-year simulation and six orbit periods at seasonal start time: (**a**) $T_x$ and (**b**) $T_y$.

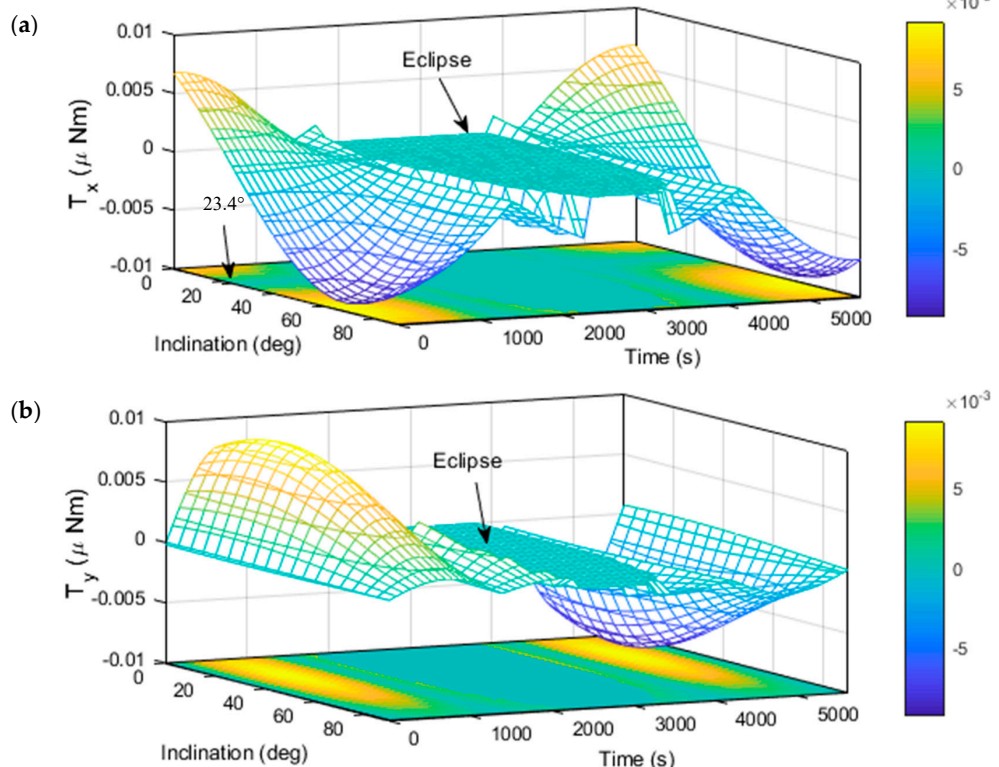

**Figure 12.** One orbit period of solar radiation torques in model 3 in ISS orbit with varying inclination angles: (**a**) $T_x$ and (**b**) $T_y$.

### 5.4. Solar Power Generation

The simulation results of the solar power generation in models 1, 2, and 3 over a one-year orbit period are presented in Figure 13. The sub-figures for the six orbits' simulation times, taken during the summer solstice, provide a more detailed plot of the generated power. Seasonal effects can be seen affecting the peak level of the power generated. As expected, models 1 and 3 have similar shapes since the solar radiation torques over the one year period are similar. For model 1, the peak power is 25.2 W and the average power is 10.4 W. Model 2 has the lowest peak power and average power at 16.26 W and 6.1 W, respectively. Model 3 has the highest peak power at 40.3 W. However, its average power, at 9.9 W, is lower than that of model 1.

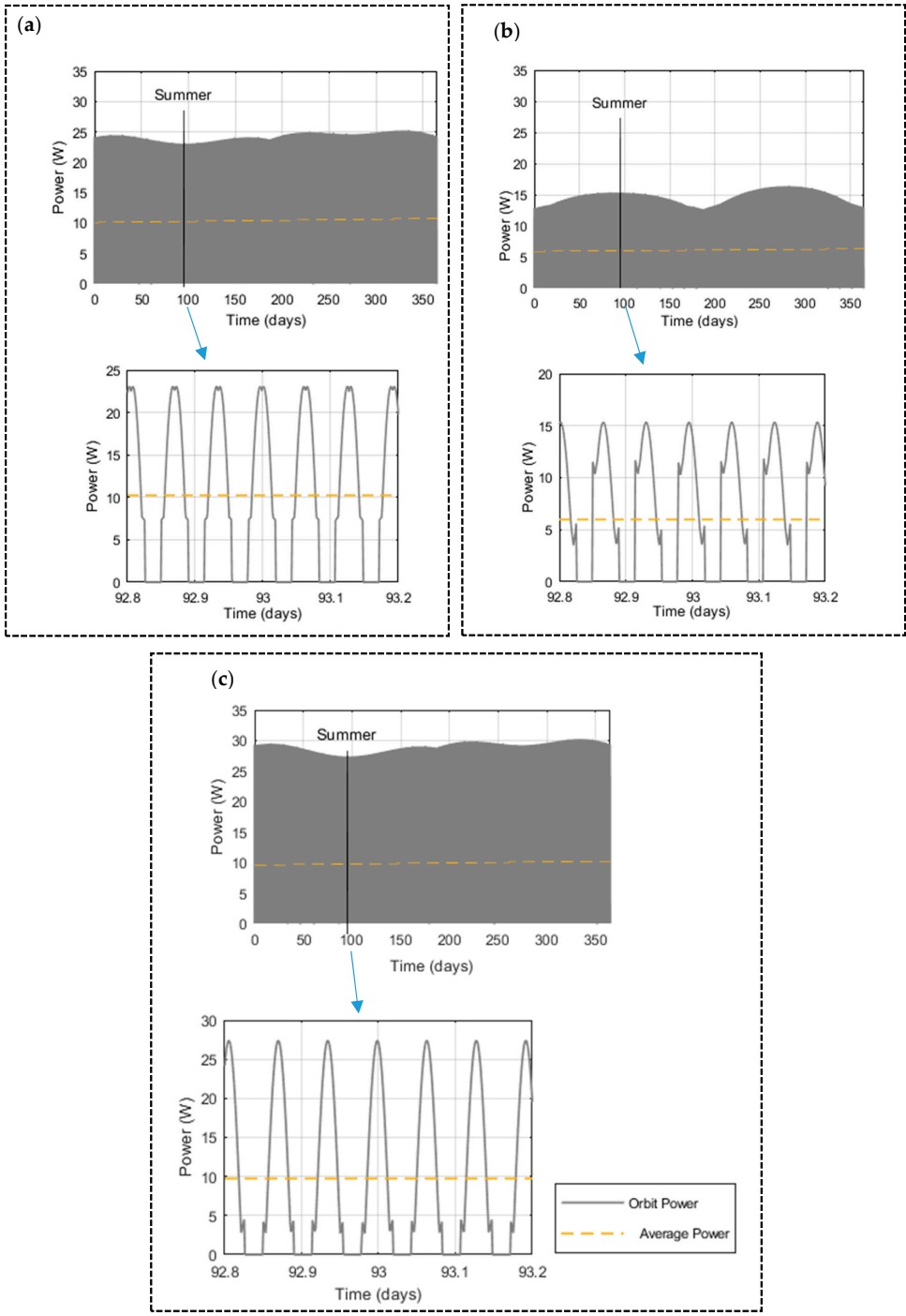

**Figure 13.** Solar power generation over one year period of (**a**) model 1, (**b**) model 2, and (**c**) model 3.

The use of deployable solar panels has significantly improved power generation, especially for models 1 and 3. Considering the same nadir-pointing missions specified, when compared with their respective configurations without deployable solar panels, model 1 performs 201% better, whereas model 3 achieves a 237% increase in average power. Model 2 produces around a 107.4% improvement.

### 5.5. Discussion—Effects of Adding Deployable Solar Panels on the Overall Disturbances

From the initial results above, the characteristics of the solar radiation torque of each CubeSat configurations were obtained. By including other external disturbances to those results, the effects of adding the deployable solar panels on the total external torques can be better observed. While the total resultant torques presented are by no means the maximum torques that will be experienced by the satellites, especially since the initial condition after deployment is not considered, forecasts on the maximum resultant torques to maintain the nadir-pointing mission proposed would be very useful for design engineers to plan the overall power usage. The one-year long period results of total torque in ISS orbit for models 1, 2, and 3 are plotted in Figures 14–16, respectively. In each figure, a comparison between the total torque with deployable solar panels and with none are made. Although not shown separately, the type of disturbance which dominates the total torque will be explained.

Firstly, the resultant $T_z$ for each model is not presented since the value is negligible compared to $T_x$ and $T_y$. For model 1, the total torques are mainly contributed by the satellite's residual dipoles. Due to the nadir-pointing mission chosen and the model's configuration, both the gravity gradient torque and the aerodynamic torque are negligible. However, the solar radiation torque contribution is still obvious, especially in the *x*-axis, as shown in Figure 14a. At the maximum, for example, during the summer solstice, the increase of disturbance torque is about 13.3% for $T_x$ and 12.7% for $T_y$ when compared to not using the solar panels.

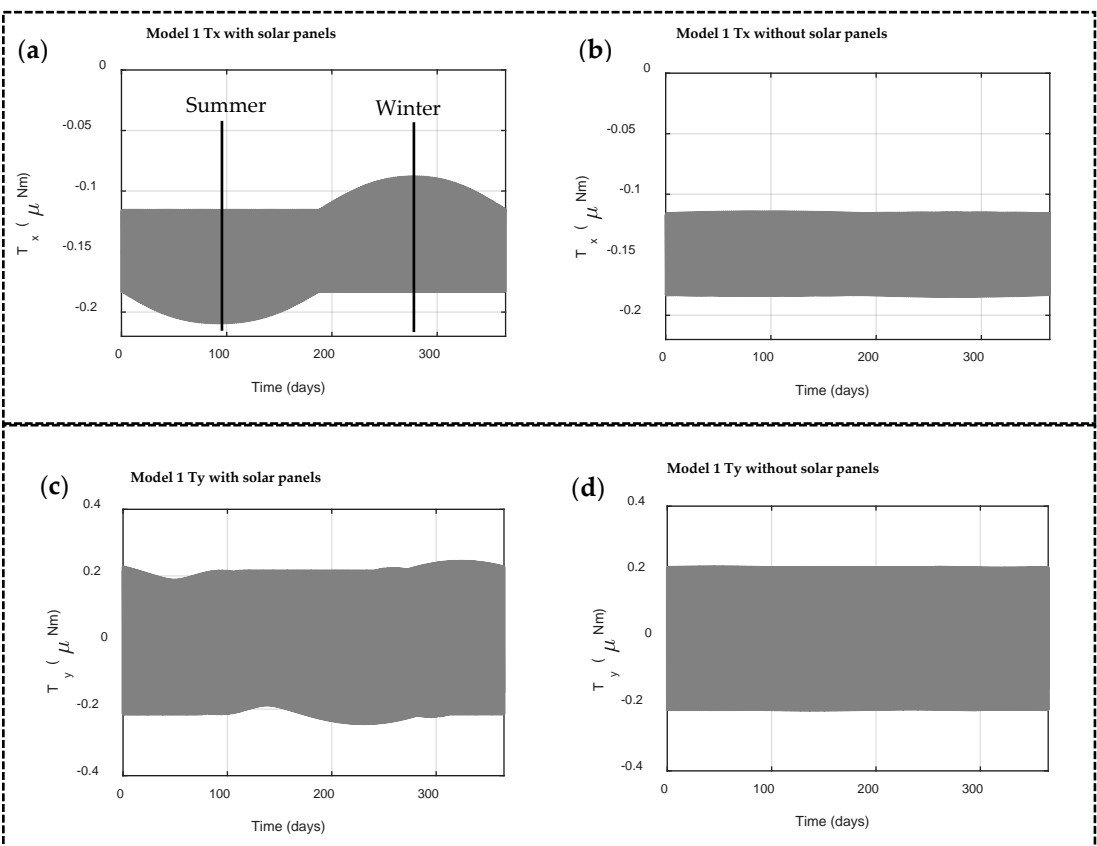

**Figure 14.** Total disturbance torque in model 1 in ISS orbit over a one year simulation period. (**a**) $T_x$ with solar panels, (**b**) $T_x$ without solar panels, (**c**) $T_y$ with solar panels, and (**d**) $T_y$ without solar panels.

Next, for model 2, a significant increase of total torque in the *x*-axis can be seen in Figure 15a when the deployable solar panels are mounted. This is mainly contributed by the aerodynamics drag as the space dart configuration contributes to a higher projected area with respect to the velocity direction. At the maximum, the increase of disturbance torque is about 322.6% for $T_x$ and 6.5% for $T_y$ when compared to not using the solar panels. Meanwhile, referring to the plot in Figure 9a, the solar radiation torque is less than 6% of the total disturbance torque.

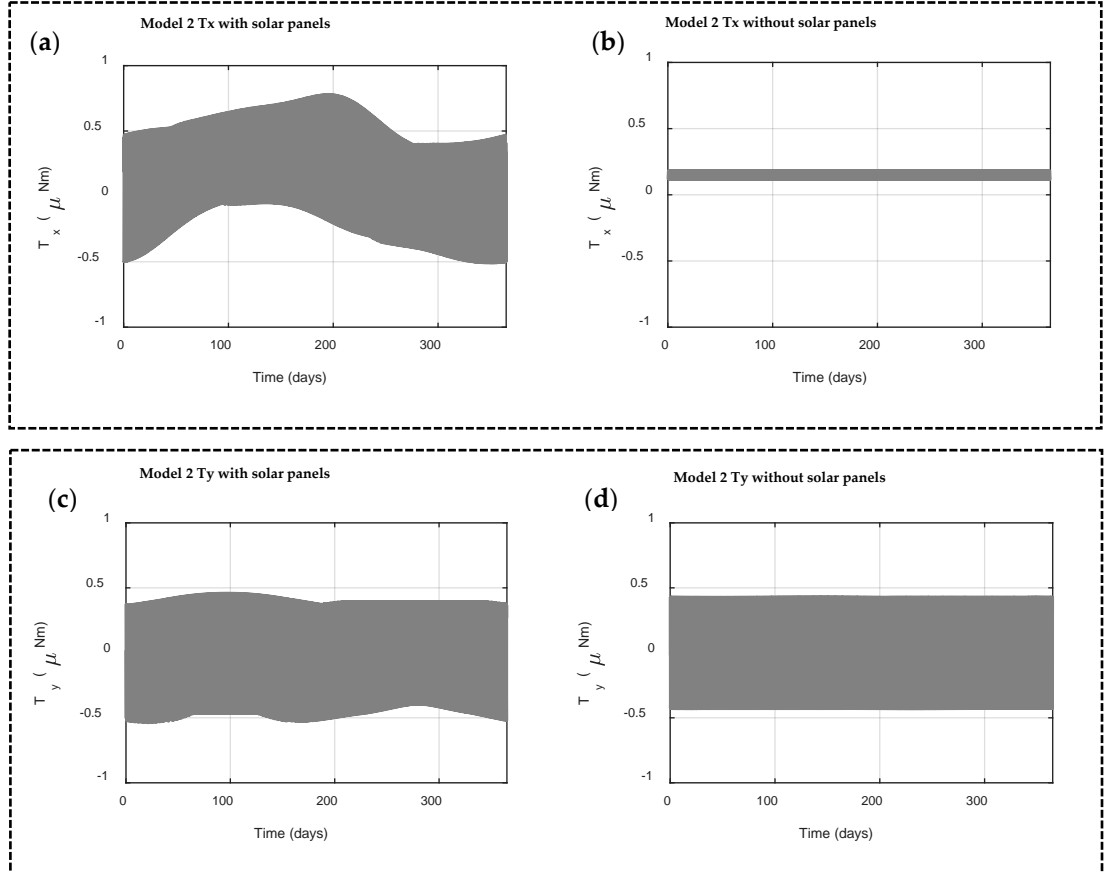

**Figure 15.** Total disturbance torque in model 2 in ISS orbit over a one year simulation period. (**a**) $T_x$ with solar panels, (**b**) $T_x$ without solar panels, (**c**) $T_y$ with solar panels, (**d**) $T_y$ without solar panels.

Lastly, for model 3, the characteristics are similar to model 1 albeit with lower total torques in both the *x*-axis and *y*-axis. As shown in Figure 16, at the maximum, the increase of disturbance torque is about 3.4% for $T_x$ and 3.2% for $T_y$ when compared to not using the solar panels.

The data obtained above show that the solar radiation torques produced due to the usage of deployable solar panels are still much smaller compared to other disturbances. The proportion should be much lower if the nadir pointing mission is not used and the satellite's center of mass is not exactly at the center of the body. On the other hand, the solar radiation torque proportion will increase when a higher altitude is used. Therefore, future work may include these variables to enable better selection of panels' configuration.

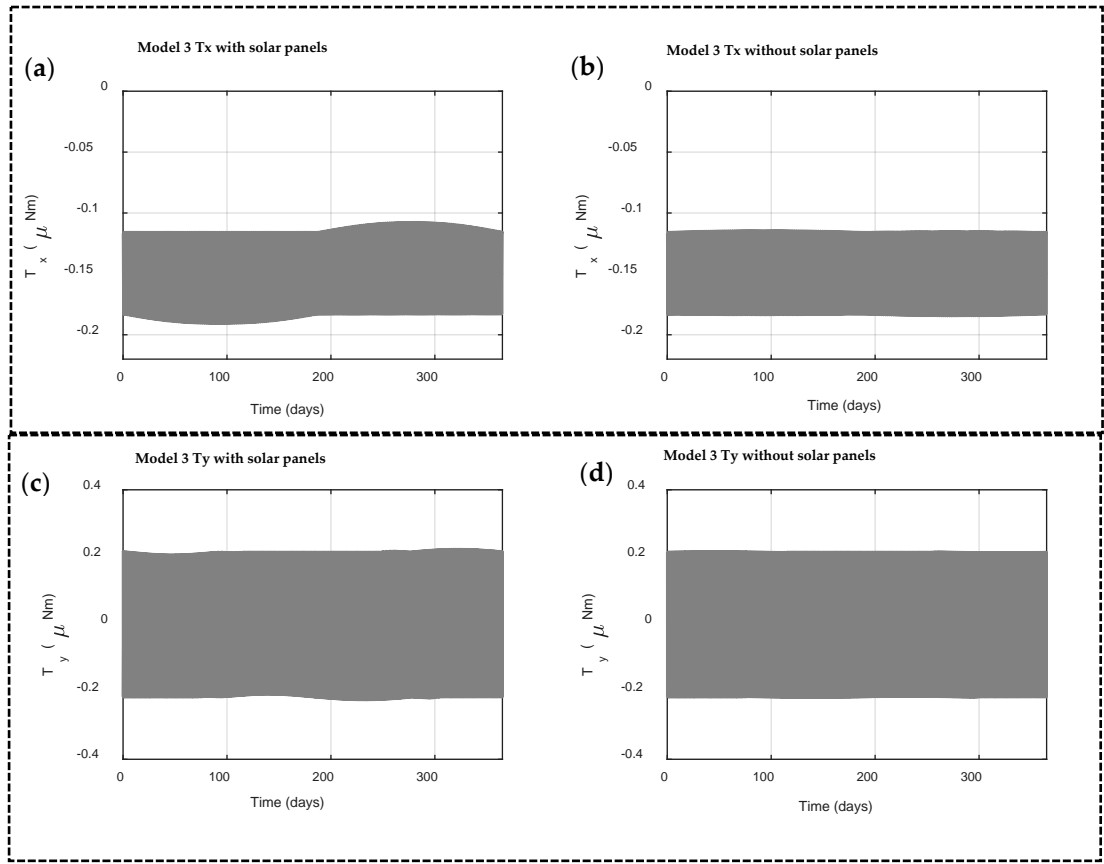

**Figure 16.** Total disturbance torque in model 3 in ISS orbit over a one year simulation period. (**a**) $T_x$ with solar panels, (**b**) $T_x$ without solar panels, (**c**) $T_y$ with solar panels, (**d**) $T_y$ without solar panels.

## 6. Conclusions

In this study, we examined the solar radiation torques encountered by low earth orbiting nanosatellites with deployable solar panels. The solar power generated when using different configurations of deployable solar panels was presented. The software algorithm used to simulate the disturbance torque and solar power generation was described in detail. The optical properties of different surfaces were distinguished and pointing missions from previous literature were considered when deciding the satellite pointing missions. For demonstration and comparison purposes, three common configurations of 3U-sized CubeSat with deployable solar panels were tested in the orbit occupied by the International Space Station. The first CubeSat has four deployable solar panels attached 90 degrees at the short edges (model 1), the second CubeSat resembles the space-dart configuration (model 2), and the third CubeSat has two-doubled solar panels deployed along the long edge of its body (model 3). With respect to the nadir-pointing missions defined, models 1 and 3 would be affected by the eclipse effect in the form of sudden torque increases and decreases. However, model 1 experiences higher peak radiation torques than model 3. For model 2, although not affected by the eclipse transition effect, the overall torque magnitudes are relatively higher compared to those in models 1 and 3. In terms of the average solar power generation, model 1 leads, followed by model 3 with just 5% less than model 1, and model 3 lags far behind, generating just 59% of the power of model 1. When compared with other external disturbances, the solar radiation torques produced are within 3.2% to 13.3% of the total torques. Model 1 and model 3 experience the residual dipole torque the most while model 2 experiences aerodynamic drag the most.

Overall, the solar radiation torque and solar power generation data presented would help CubeSats developers to select the right deployable solar panel configuration for their respective missions. Based on two factors, solar radiation torque and solar power generation, the model 2 configuration would be

the best option for nadir-pointing missions. Future work will extend the impact of the solar radiation torque on the dynamics of the respective CubeSat models used in the study, as well as the required control torque to maintain the proposed nadir pointing mission.

**Author Contributions:** Analysis, software, and writing—original draft preparation, S.A.I.; supervision and writing—review and editing, E.Y.

**Funding:** This research was funded by Jabatan Perkhidmatan Awam Malaysia (HLP2015).

**Acknowledgments:** The authors gratefully acknowledge the support from the National Space Agency of Malaysia (ANGKASA) in conducting this research.

**Conflicts of Interest:** The authors declare no conflict of interest.

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
