# Peer review of "Comparison of Solar Radiation Torque and Power Generation of Deployable Solar Panel Configurations on Nanosatellites"

_aerospace, doi:10.3390/aerospace6050050_

Round 1
Reviewer 1 Report
Summary
This work compares three different 3U CubeSat deployable panels configurations in function of the solar radiation torques and power production. A detailed model for radiation pressure is presented and a simulation framework is introduced. A second part of the paper focuses on the three configurations behaviour: for each one, torques are calculated for one year of operations and varying inclination, at a fixed altitude of about 400 km.
Broad comments
The paper is well written and well organized. A clear introduction describes the problem and how it is addressed in the paper. The first parts, dedicated to radiation pressure and astrodynamics models, is highly detailed (see dedicated comments). Simulation results are clearly presented.
I warmly recommend adding a short section before conclusions to comment the simulation results. With a couple of paragraphs it would be possible to summarize the main findings and perform a more structured comparison among the proposed configurations. In particular, it would be interesting to introduce some comparison parameters such as the ratio between average or peak torques and average produced power, to better define the configuration performance. A further advancement would be to compare the radiation pressure torques with other disturbances, to better understand the required control torque to maintain the proposed attitude: another interesting parameter would be the ratio between average control torque and produced power.
In general, the quality of this work is good. I would conclude that this paper would require only minor reviews. In the next list, some other minor suggestions/opinions:
1. There is no detail in the paper on view factors: are they automatically calculated by the simulation software? They are fundamental in defining shadow effects and the contribution of albedo and Earth radiation.
2. On page 4, line 115, panels thickness is estimated in 30 mm with no references. This value seems particularly high, considering CubeSat size; I wonder if values around 3-5 mm would be more plausible.
3. Page 6, line 190, I cannot find where the coefficient rho_p is introduced. Is might be a typo with rho_E.
4. Page 6, line 194, the solar cell number (on sigma) is fixed at 16, but only the third configuration shows 16 cells; please check it.
5. On the same line, I wonder if, for completeness, the scalar product could be substituted with a mathematical expression stating the concept introduced in the following paragraph, e.g.:
max(0, s^T n)
6. Page 9, in the calculation of surface normal, the text gave me at first the impression that the albedo and the Earth radiation forces are calculated only for surfaces not facing the sun. Please check the test to avoid any potential misunderstanding.
Author Response
Response to Reviewer 1 Comments
We appreciate so much for all the constructive comments from the reviewer. We also thank the reviewer for the effort and time put into the review of the manuscript. We have tried to consider each comment and respond to them. Responses to the reviewer and changes in the revised manuscript are as follows.
Point 1: I warmly recommend adding a short section before conclusions to comment the simulation results. With a couple of paragraphs it would be possible to summarize the main findings and perform a more structured comparison among the proposed configurations. In particular, it would be interesting to introduce some comparison parameters such as the ratio between average or peak torques and average produced power, to better define the configuration performance. A further advancement would be to compare the radiation pressure torques with other disturbances, to better understand the required control torque to maintain the proposed attitude: another interesting parameter would be the ratio between average control torque and produced power.
Response 1: Thank you for your recommendations. Based on your suggestions, we think that comparison with other disturbances would be useful in giving readers foresight on the additional increase of disturbance torque when using the deployable solar panels in the CubeSat design. Therefore, we added brief information on the models used to compute other external disturbances and used the results to discuss the effects of using deployable solar panels onto the total disturbances face by the CubeSats. In addition, we highlighted the subject of control torque as a future work in the conclusion section.
Correction on Page 3, Line 92:
Added section:
Section 2.2. Other Disturbances.
Correction on Page 21, Line 566:
Added section:
Section 5.5. Discussion – effects of adding deployable solar panels on the overall disturbances.
Correction on Page 23, Line 659:
Added sentence:
“Future work will extend the impact of the solar radiation torque on the dynamics of the respective CubeSat models used in the study, as well as the required control torque to maintain the proposed nadir pointing mission. “
Point 2: There is no detail in the paper on view factors: are they automatically calculated by the simulation software? They are fundamental in defining shadow effects and the contribution of albedo and Earth radiation.
Response 2: Thank you for your comment. There is no detail in the paper on view factors because we used the sun vector and satellite vector to measure the contribution of albedo and Earth radiation. In Page 2, line 67, we assumed that Earth radiation is uniform over the Earth surface. In the measurement, we used satellite vector to measure the force due to Earth radiation. In Page 2, line 74, we assumed the reflection from the earth’s surface and from clouds as diffuse. The flux from the total reflectance can then be computed by integrating over the surface of the earth. In the measurement (Equation 5), we used both sun vector and satellite vector to measure the force due to albedo. We also revised Figure 6 by adding Earth radiation and radiation due to albedo and Earth to make them clear to readers.
Correction on Page 2, Line 74:
Added sentences:
“The flux due to albedo can then be computed by integrating over the surface of the earth.”
Correction on Page 10, Line 372:
Added sentences:
“Accordingly, the fraction of solar flux follows suit (i.e., occurs only during the daylight period). As shown in Figure 6, the Earth-emitted radiation is constant throughout the orbit, since we previously assumed that it is uniform over the surface of the earth, whereas the reflected radiation due to albedo varies with the sun vector and satellite vector from the Earth.”
Correction on Page 11, Line 381:
We revised Figure 6 by adding data of Earth radiation and reflected radiation due to albedo.
Point 3: On page 4, line 115, panels thickness is estimated in 30 mm with no references. This value seems particularly high, considering CubeSat size; I wonder if values around 3-5 mm would be more plausible.
Response 3: Thank you for highlighting the mistake. The panel thickness should be set at 3 mm, instead of 30 mm. The correction is made in Page 4, line 144.
Point 4: Page 6, line 190, I cannot find where the coefficient rho_p is introduced. Is might be a typo with rho_E.
Response 4: Thank you for highlighting the mistake. The coefficient should be rho_E, instead of rho_p. To avoid confusion with rho symbols used to describe properties, we replaced the rho symbols used in Equations 13 to 16, and Figure 3 to psi symbols. The corrections are made in Page 6, Figure 3 and Page 6, Line 212-220.
Point 5: Page 6, line 194, the solar cell number (on sigma) is fixed at 16, but only the third configuration shows 16 cells; please check it.
Response 5: Thank you for your comment. We think that we have explained in Page 4, lines 144 to 146 that there is a total of 30 surfaces. Then in Page 4 and Page 5, lines 153 to 168, we explained which surfaces are assigned with solar cell surface properties and radiator surface properties. Therefore, in the end, each satellite model has a total of 16 surfaces with solar cell surface properties, hence the fixed number of 16 cells in the Equation 17 (Page 6, line 224).
Point 6: On the same line, I wonder if, for completeness, the scalar product could be substituted with a mathematical expression stating the concept introduced in the following paragraph, e.g.: max(0, s^T n)
Response 6: Thank you for your comment. We added “for (s^T n) > 0” in the Equation 17 (Page 6, line 224).
Point 7: Page 9, in the calculation of surface normal, the text gave me at first the impression that the albedo and the Earth radiation forces are calculated only for surfaces not facing the sun. Please check the test to avoid any potential misunderstanding.
Response 7: Thank you for your comment. Yes, the albedo and the Earth radiation forces are calculated only for surfaces not facing the sun. To better explain this matter, we added an explanation for Equation 5 and Equation 24 as follows:
i. Page 2, line 85 – added “for (s^T n)> 0”.
ii. Page 3, line 88 – added “The s^T n is the dot product, which is the cosine of the angle between s^ and n. Its positive value means that the surface normal faces to the sun direction. Later, we explain that the s^ value is also interchangeable with the other two sources of radiation which are the Earth’s radiation and albedo.”
iii. Page 9, line 347 – added “The negative sign in Equation (27) indicates that the source of radiation is from the nadir direction.”
END

Reviewer 2 Report
The paper deals with the analysis of the impact of solar radiation on the attitude of a small satellite with different configuration of deployed solar panels.
The introduction and the analysis of the state of the art are sufficient and the problem is described.
Methodology and models description are well structured. Moreover, it is not clear what is the relationship between the generated power and the disturbance (assuming that a relationship exists)
Results are of interest but a discussion about the impact of the radiation torques on the dynamics of the satellite, i.e. for which operative modes (e.g. fine pointing) and/or kinds of maneuver (quick/slow slew maneuvers) this torque can affect the operativity of the entire ADCS. In particular, a budget and relative comparison among the major disturbance torque and the three different configuration should be included.
Moreover, it should be highlighted what are the case studies of missions for which the analysis has been led. Please highlight/summarize in a specific (sub-) paragraph all the assumptions and discuss the range of applicability of the results. At the moment, not all the initial conditions and the parameters of the simulations are clear or well defined.
Conclusions are the summary of the paper. Please provide few sentences about the perspective of the works.
Finally, English language requires some review.
Author Response
Response to Reviewer 2 Comments
We appreciate so much for all the constructive comments from the reviewer. We also thank the reviewer for the effort and time put into the review of the manuscript. We have tried to consider each comment and respond to them. Responses to the reviewer and changes in the revised manuscript are as follows.
Point 1: Methodology and models description are well structured. Moreover, it is not clear what is the relationship between the generated power and the disturbance (assuming that a relationship exists)
Response 1: Thank you for your comment. The relation is mainly due to the larger area used to mount the solar arrays. In addition, the way the panels are configured also contributing to differences in the resultant solar radiation torque patterns. However, due to your comment, we think that it is better to change the title of the manuscript to make it clear to readers. Therefore, we would like to change the title to “Comparison Study on Solar Radiation Torque and Power Generation of Deployable Solar Panels Configurations of Nanosatellite”
Point 2: Results are of interest but a discussion about the impact of the radiation torques on the dynamics of the satellite, i.e. for which operative modes (e.g. fine pointing) and/or kinds of maneuver (quick/slow slew maneuvers) this torque can affect the operativity of the entire ADCS. In particular, a budget and relative comparison among the major disturbance torque and the three different configurations should be included.
Response 2: Thank you for your comment. We do not cover the ADCS part since it is not the focus of this study as yet. Nevertheless, we briefly addressed this matter as the future work that we plan to do in the conclusion part, so that general readers can recognize the parts where this study should be extended. Then, based on your suggestions, we think that comparison with other disturbances would be useful in giving readers foresight on the additional increase of disturbance torque when using the deployable solar panels in the CubeSat design. Therefore, we added brief information on the models used to compute other external disturbances and used the results to discuss the effects of using deployable solar panels onto the total disturbances face by the CubeSats.
Correction on Page 3, Line 92:
Added section:
Section 2.2. Other Disturbances.
Correction on Page 21, Line 566:
Added section:
Section 5.5. Discussion – effects of adding deployable solar panels on the overall disturbances.
Correction on Page 23, Line 659:
Added sentence:
“Future work will extend the impact of the solar radiation torque on the dynamics of the respective CubeSat models used in the study, as well as the required control torque to maintain the proposed nadir pointing mission. “
Point 3: Moreover, it should be highlighted what are the case studies of missions for which the analysis has been led.
Response 3: Thank you for your comment. For this matter, we do not want to be too specific since we want to make it easier for comparison. Therefore, we decided to use the same mission; nadir pointing mission for the three models used, as mentioned in Section 3.1. However, we think that it is not written clearly. Hence, we added the following sentences, before elaborating which part of the CubeSats that faces to the nadir.
Correction on Page 4, Line 125:
Added sentences:
For ease of comparison, all the CubeSat models used will have the same mission that is a nadir-pointing mission. It is assumed that their respective attitude control systems are able to maintain the pointing, in the direction assigned. The current state-of-the-art of CubeSat technology has been reviewed to be potentially compatible with some Earth observation missions [17].
Point 4: Please highlight/summarize in a specific (sub-) paragraph all the assumptions and discuss the range of applicability of the results. At the moment, not all the initial conditions and the parameters of the simulations are clear or well defined.
Response 4: Thank you for your comment. Regarding the initial conditions (i.e. Table 1), we think that we have explained the important initial conditions in paragraph 2 of Section 4 (Page 8, line 301). We mentioned in the paragraph the reasons of the chosen ISS orbit, as well as variation applied on some Kepler elements. As for the parameters in Table 2, we add a reference to the surface properties used. The rest of the parameters have been explained in previous sections.
Point 5: Conclusions are the summary of the paper. Please provide few sentences about the perspective of the works.
Response 5: Thank you for your comment. Regarding the perspective of the work, we used the results of total torques caused by the deployable solar panels as a summary for the overall study. That is the discussion part on the total disturbance torque (same as point 2 above).
Point 6: Finally, English language requires some review.
Response 6: English was reviewed and corrected.
END

Round 2
Reviewer 2 Report
Major issues of the previous review have been addressed. I appreciated the effort of the authors both in the improvements in the text and in the answers to my questions and requests of clarification. I think that the actual version remains actually general for some real aspects but I consider that the level reached by this version of the paper is satisfying for the publication.